# Minimization of Inhibitor Generation in Rice Straw Hydrolysate Using RSM Optimization Technique

**Vikas Chandra Gupta** [1,*], **Meenu Singh** [1], **Shiv Prasad** [2,*] and **Bhartendu Nath Mishra** [3]

1   Department of Biotechnology, College of Engineering and Technology, IILM University, Greater Noida 201310, India; meenu.singh@iilm.edu
2   Division of Environment Science, ICAR-Indian Agricultural Research Institute (IARI), New Delhi 110012, India
3   Department of Biotechnology, Institute of Engineering and Technology, Dr. A.P.J. Abdul Kalam Technical University, Lucknow 226021, India; bnmishra@ietlucknw.ac.in
*   Correspondence: vikas.chandragupta01@gmail.com (V.C.G.); shiv_drprasad@yahoo.co.in (S.P.)

**Abstract:** Ethanol production from lignocellulosic biomass comprises pretreatment, hydrolysis, and fermentation. However, several inhibitors are generated during rice straw chemical hydrolysis, including furfural, 5-hydroxymethylfurfural (HMF), and phenolics. These inhibitors, i.e., furfural and HMF, are toxic to yeast cells, can negatively impact yeast growth and metabolism, and reduce the process efficiency and production yield. Total phenolics are also reported to inhibit yeast growth and metabolism and act as a source of reactive oxygen species (ROS), which can damage yeast cells. Therefore, minimizing the generation of these inhibitors during rice straw hydrolysis is essential to improve the efficiency and yield of ethanol fermentation. Optimization of process variables can help reduce inhibitor generation and increase the efficiency of used detoxification methods such as adsorption, ion exchange, and biological methods. This study aimed to minimize inhibitor generation during the chemical hydrolysis of rice straw biomass. Minitab 17 software was employed and response surface curve regression analysis was used to develop a quadratic equation of an optimized process for minimized release of inhibitors molecules. The main inhibitors in pretreated rice straw hydrolysate identified were furfural (48.60%/100 g solid biomass), HMF (2.32%/100 g solid biomass), and total phenolics (1.65%/100 g solid biomass). The optimal pretreatment conditions were a biomass solid loading rate of 15% $w/v$, an $H_2SO_4$ concentration of 12% $v/v$, a pretreatment reaction time of 30 min, and a temperature of 100 °C. Optimization of these process variables reduced the inhibitor generation by up to one and a half fold.

**Keywords:** rice straw; pretreatment; inhibitors; response minimization; optimized yield





## 1. Introduction

The utilization of lignocellulosic biomass for biofuel production has gained much attention in recent years due to its potential as a renewable and sustainable alternative to traditional fossil fuels [1]. Among the various lignocellulosic biomass sources, rice straw is an abundant and low-cost agricultural waste that can be utilized to produce biofuels and other high-value products [2]. However, the crystalline compositional structure of biomass and the process-based generation of inhibitors molecules such as furfural and 5-hydroxymethylfurfural (HMF) in rice straw hydrolysate can significantly reduce the efficiency of the hydrolysis process [3]. Several approaches have been proposed to minimize the generation of inhibitors in rice straw hydrolysate [4].

Physical methods, such as steam explosion and liquid hot water pretreatment, have been shown to reduce the concentration of inhibitors in the hydrolysate [5]. However, these methods require high energy consumption and can be expensive [6]. Chemical methods, such as acid and alkaline pretreatment, have also been used to reduce the concentration of inhibitors in the hydrolysate [7]. However, these methods have limitations in terms

of cost, environmental impact, and efficiency [8]. Biological methods, such as the use of inhibitor-tolerant microorganisms and genetic engineering, have also been proposed to overcome the inhibition problem [9]. However, these methods are expensive, and research and development can be highly time-consuming [10].

During the pretreatment of rice straw, high temperatures and acid concentrations lead to the production of inhibitors such as pentose-derived furfurals, hexose-derived hydroxymethylfurfural (HMF), total phenolics, and various acids like acetic and formic acid [11]. These inhibitors must be controlled and reduced to a minimum level during the process to achieve the sustainable production of precursor fermentable sugar molecules for ethanol production. The factors responsible for inhibitor production include reaction temperature, acid concentration, autoclave time, and solid biomass concentration [12]. Studies have shown that pretreatment of solid biomass with high acid and temperature generates higher concentrations of degradation products like furfural and HMF, with an improved chance of fermentable sugar yield from the process [13]. Aromatic degradation products like furans and phenols also form due to sugar degradation during this process [14].

The type of pretreatment and the lignin ratio in the biomass material affect the concentration of these compounds in hydrolysates [15]. In order to optimize and reduce the generation of such inhibitor molecules from the rice straw hydrolysis process, mathematical and statistical analysis tools, such as the factorial design of experiments and response surface methodology, have been employed [16]. The response surface methodology (RSM) optimization technique has been successfully used in various fields to optimize process variables and minimize the occurrence of unwanted by-products [17]. In recent years, RSM has been applied to optimize the bioconversion process of lignocellulosic biomass to bioethanol and other high-value products [18]. RSM can optimize the process variables such as temperature, reaction time, and acid concentration to minimize the concentration of inhibitors in the hydrolysate, leading to increased bioconversion efficiency [19].

Studies have shown that the RSM optimization technique can reduce inhibitor generation in rice straw hydrolysate [20–22]. This technique was used to refine the hydrolysis and pretreatment conditions of rice straw, leading to lower furfural and HMF concentrations in the hydrolysate and improved bio-conversion efficiency [23–27]. The RSM optimization technique has emerged as an effective and sustainable approach to minimizing inhibitor generation in rice straw hydrolysate. The optimization of process variables such as temperature, reaction time, and acid concentration can significantly reduce the concentration of furfural, HMF, and total phenolics in the hydrolysate, leading to increased bioconversion efficiency. The application of the RSM optimization technique in rice straw bioconversion can contribute to the development of a sustainable and efficient biofuel production system. The present research study aimed to optimize key process variables to reduce the inhibitor generation in the rice straw hydrolysate with improved process efficiency.

## 2. Materials and Methods

### 2.1. Rice Straw Preparation and Pretreatment Optimization Using CCD and RSM Methods

Response surface methodology (RSM) was used to optimize acidic pretreatment by studying the effects of four variables on the process efficiency with minimized inhibitors release [28]. The washed, dried, and powdered rice straw substrate of 30–50 mm size was pretreated using a combination of pretreatment variables. These variables were (A) biomass solid loading, (B) $H_2SO_4$ concentration, (C) reaction time, and (D) temperature.

The study employed the central composite design (CCD) of experiments to construct a mathematical model that established connections between the variables and the optimized outcomes in the hydrolysate. These outcomes included minimizing the release of furfural, HMF, and total phenolics while simultaneously maximizing the release of fermentable sugars. The CCD involved conducting a total of 31 experiments, and the resulting quadratic equation effectively captured the response of the experimental design. This equation incorporated the four independent variables and their interactions, with the response being the minimized quantities of furfural, HMF, and total phenolics, as well as

the maximized amount of reducing sugars released. The following quadratic equation explains the response of experimental design:

$$Y = \beta_0 + \beta_1 A + \beta_2 B + \beta_3 C + \beta_4 D + \beta_{11} A_2 + \beta_{22} B_2 + \beta_{33} C_2 + \beta_{44} D_2 + \beta_1\beta_2 AB + \beta_1\beta_3 AC + \beta_1\beta_4 AD + \beta_2\beta_3\beta_4 BCD,$$

where Y is the optimized process response; $\beta$ is the factor coefficient; A, B, C, and D are the biomass solid loading (%*w/v*), $H_2SO_4$ (%*v/v*), reaction time (min), and temperature (°C), respectively.

### 2.2. Quantification of Furfural and Hydroxymethylfurfural

Carbohydrate-derived inhibitor molecules, namely furans, were detected and quantified using the supernatant of pretreated rice straw obtained from chemical pretreatment steps. The UV-spectrophotometer method for furfural and HMF analysis was performed [29]. The acid-pretreated rice straw hydrolysate was filtered through a Whatman grade No. 1 filter paper to separate the solid residues and liquid, and the filtrate fraction was collected and centrifuged at 0 °C at 10,000 rpm for 10 min. It was then stored at 4 °C in the refrigerator for quantifying the furan concentration. For this, 5 mL of clear filtrate from the pretreatment step was taken in a 25 mL volumetric flask, and 8.5 mL of concentrated hydrochloric acid and 7 mL of 3XM TBA (2-thiobarbituric acid) solution were added. The samples were then heated to 40 °C for 30 min and allowed to cool to 20 °C. The volume was made up using double-distilled water. The absorbance spectrum was recorded at 436 nm for furfural and 414 nm for HMF, and each concentration was calculated based on the standard graphs of the same compounds.

### 2.3. Quantification of Total Phenolics Compounds

The lignin-based derivatives, as total phenolics concentration, were estimated using Folin–Ciocalteu's colorimetric method for phenolics compounds estimation [30]. After centrifugation at 10,000 rpm for 10 min, the filtrate obtained from pretreated rice straw hydrolysate was used for measuring the phenolics content. For this, 50 μL of the clear filtrate of rice straw hydrolysate was added to 950 μL of double-distilled water and 1500 μL of the Folin-phenol solution in the test tube. After 3 min, 2000 μL of sodium bicarbonate (10%, *w/v*) was added to the mixture. The absorbance was measured at 765 nm after the 4-h incubation period in the darkroom. Gallic acid was used for the standard calibration curve.

## 3. Results

### 3.1. Statistical Analysis of the Minimized Release of Furfural (%) in Pretreated Rice Straw Hydrolysate

The parameters used to obtain a higher yield of reducing sugar in rice straw hydrolysate have resulted in the formation of potential inhibitor molecules as degradation by-products during the pretreatment of rice straw. These potential inhibitors, namely furfural (48.60%/100 g of solid biomass), 5-hydroxymethylfurfural (HMF) (2.32%/100 g of solid biomass), and total phenolics (1.65%/100 g of solid biomass), were obtained in varying concentrations based on the treatment conditions presented in Table 1. The high temperature and acid concentration used in the pretreatment process seems to have generated these potential inhibitor molecules, though in reduced and minimized concentrations due to process variables screening and optimization of rice straw chemical hydrolysis. The pretreatment condition that resulted in the reduced concentrations of potential inhibitor molecules with improved process efficiency involved a biomass solid loading rate of 15% *w/v*, an $H_2SO_4$ concentration of 12% *v/v*, a reaction time of 30 min, and a temperature of 100 °C.

The regression equation for furfural (%) from the experimental trials is given below for the computation of furfural estimates:

$$\begin{aligned} \text{furfural (\%)} = &-84.43 - 0.060A + 1.512B + 2.105C + 1.3850D + 0.00051A \times A - 0.13156 \\ &B \times B - 0.02471C \times C - 0.003731D \times D + 0.03311A \times B + 0.06913A \times C - 0.01419A \times D - \\ &0.00517B \times C + 0.00944B \times D - 0.014194 C \times D, \end{aligned} \quad (1)$$

where Equation (1) predicts the yield of furfural (in %), in which A, B, C, and D are the coded values of biomass solid loading ($\%w/v$), $H_2SO_4$ ($\%v/v$), reaction time (min), and temperature (°C), respectively.

**Table 1.** Inhibitors profile generated from pretreatment of rice straw.

| Std Order | Run Order | Pt Type | Blocks | Biomass Solid Loading (%w/v) | H₂SO₄ (%v/v) | Reaction Time (min) | Temp (°C) | Reducing Sugar (mg/g Biomass) * | Furfural (%) * | Hmf (%) * | Total Phenolics (%) * |
|---|---|---|---|---|---|---|---|---|---|---|---|
| 11 | 1 | 1 | 1 | 10 | 20 | 35 | 170 | 326.5 | 6.94 | 1.07 | 1.058 |
| 27 | 2 | 0 | 1 | 17.5 | 12.5 | 25 | 125 | 273.5 | 7.64 | 1.19 | 1.043 |
| 24 | 3 | −1 | 1 | 10 | 5 | 15 | 80 | 168.4 | 8.85 | 0.40 | 1.104 |
| 8 | 4 | 1 | 1 | 10 | 5 | 35 | 80 | 123.3 | 12.85 | 0.73 | 0.478 |
| 3 | 5 | 1 | 1 | 17.5 | 27.5 | 25 | 125 | 153.1 | 13.75 | 0.58 | 0.835 |
| 26 | 6 | 0 | 1 | 17.5 | 12.5 | 25 | 125 | 279.6 | 14.07 | 1.30 | 0.883 |
| 31 | 7 | 0 | 1 | 17.5 | 12.5 | 25 | 215 | 219.8 | 14.29 | 0.59 | 0.93 |
| 10 | 8 | 1 | 1 | 17.5 | 12.5 | 25 | 125 | 283.4 | 14.95 | 0.17 | 1.136 |
| 20 | 9 | −1 | 1 | 25 | 5 | 35 | 170 | 164.5 | 15.47 | 1.15 | 1.02 |
| 23 | 10 | −1 | 1 | 17.5 | 2.5 | 25 | 125 | 183 | 16.92 | 0.57 | 1.011 |
| 29 | 11 | 0 | 1 | 17.5 | 12.5 | 25 | 125 | 286.1 | 19.5 | 1.42 | 0.821 |
| 4 | 12 | 1 | 1 | 32.5 | 12.5 | 25 | 125 | 203.7 | 19.88 | 0.38 | 1.496 |
| 17 | 13 | −1 | 1 | 10 | 20 | 15 | 170 | 348.2 | 22.09 | 2.32 | 0.76 |
| 15 | 14 | 1 | 1 | 17.5 | 12.5 | 5 | 125 | 225.2 | 23.71 | 0.03 | 0.97 |
| 1 | 15 | 1 | 1 | 2.5 | 12.5 | 25 | 125 | 221.3 | 25.43 | 2.12 | 1.21 |
| 9 | 16 | 1 | 1 | 25 | 20 | 15 | 80 | 132.8 | 27.06 | 0.03 | 0.541 |
| 25 | 17 | 0 | 1 | 10 | 20 | 15 | 80 | 212.7 | 27.48 | 0.51 | 0.754 |
| 7 | 18 | 1 | 1 | 25 | 20 | 35 | 170 | 193.6 | 32.52 | 0.29 | 1.501 |
| 16 | 19 | 1 | 1 | 10 | 5 | 35 | 170 | 101.2 | 33.41 | 0.68 | 0.749 |
| 28 | 20 | 0 | 1 | 17.5 | 12.5 | 25 | 125 | 293.4 | 35.03 | 1.19 | 1.647 |
| 2 | 21 | 1 | 1 | 10 | 20 | 35 | 80 | 232.8 | 35.33 | 0.92 | 1.32 |
| 6 | 22 | 1 | 1 | 25 | 5 | 15 | 80 | 238.5 | 36.57 | 1.13 | 0.63 |
| 12 | 23 | 1 | 1 | 25 | 5 | 15 | 170 | 200.5 | 38.43 | 0.41 | 1.052 |
| 5 | 24 | 1 | 1 | 25 | 20 | 15 | 170 | 165.4 | 39.34 | 0.03 | 0.91 |
| 14 | 25 | 1 | 1 | 25 | 20 | 35 | 80 | 184.5 | 41.75 | 1.17 | 0.986 |
| 21 | 26 | −1 | 1 | 17.5 | 12.5 | 45 | 125 | 208.3 | 41.94 | 1.25 | 1.013 |
| 13 | 27 | 1 | 1 | 10 | 5 | 15 | 170 | 169.9 | 42.03 | 1.46 | 0.91 |
| 22 | 28 | −1 | 1 | 25 | 5 | 35 | 80 | 225 | 42.59 | 1.74 | 0.683 |
| 18 | 29 | −1 | 1 | 17.5 | 12.5 | 25 | 35 | 209.3 | 42.64 | 1.00 | 0.594 |
| 19 | 30 | −1 | 1 | 17.5 | 12.5 | 25 | 125 | 301.2 | 45.13 | 1.28 | 1.224 |
| 30 | 31 | 0 | 1 | 17.5 | 12.5 | 25 | 125 | 306.8 | 48.6 | 1.03 | 1.268 |

* Response was measured in triplicate, and the mean data are shown in the table.

The response surface curve and interaction between different process variables are shown in Figure 1a–f. The minimized furfural release in the pretreated rice straw hydrolysate was found to be in the range of, i.e., 6.94–48.6% furfural/100 g of solid pretreated biomass (Table 1). The values of each variable found at the optimum operating level were selected as (A) biomass solid loading (15% $w/v$), (B) $H_2SO_4$ concentration (12% $v/v$), (C) reaction time of pretreatment (30 min), and (D) temperature found to be optimum at (100 °C).

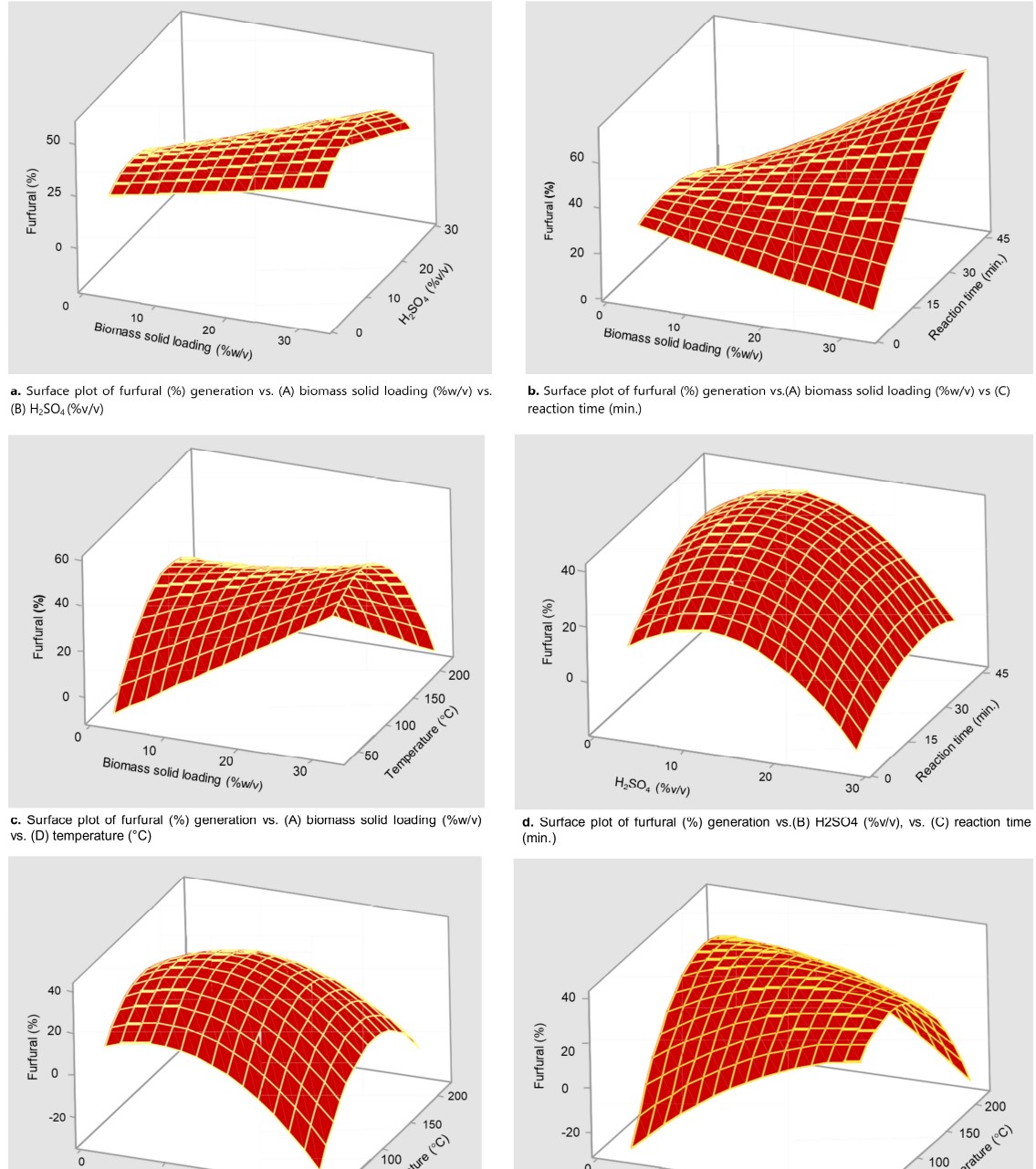

**Figure 1.** (**a**–**f**): The response surface curve and interaction between different process variables for furfural generation.

In Figure 1a, a surface plot is presented that shows the generation of furfural (%) in relation to two variables: (A) biomass solid loading (15% *w/v*) and (B) H$_2$SO$_4$ concentration (12% *v/v*). The furfural obtained in the hydrolysis process may be due to the high acid concentration used in the study to hydrolyze the rice straw solid biomass effectively and readily. The significance of the interaction has been validated statistically with an F-value of 25.74 and a Prob > F of less than 0.00001 (Table 2). A similar interaction plot shows that the factor responsible for minimizing the release of fermentation inhibitors has an incremental impact on rice straw pretreatment using high acid and temperature.

**Table 2.** Response surface regression: furfural (%) versus (A) biomass solid loading (%$w/v$), (B) H$_2$SO$_4$ (%$v/v$), (C) reaction time (min), (D) temp (°C).

| Source | DF | Seq SS | Contribution | Adj SS | Adj MS | F-Value | *p*-Value |
|---|---|---|---|---|---|---|---|
| **Analysis of Variance** | | | | | | | |
| Model | 14 | 4827.76 | 99.29% | 4827.76 | 344.84 | 159.91 | 0.000 |
| Linear | 4 | 483.23 | 9.94% | 978.83 | 244.71 | 113.47 | 0.000 |
| A | 1 | 144.06 | 2.96% | 619.34 | 619.34 | 287.20 | 0.000 |
| B | 1 | 148.54 | 3.06% | 76.21 | 76.21 | 35.34 | 0.000 |
| C | 1 | 138.24 | 2.84% | 30.63 | 30.63 | 14.20 | 0.002 |
| D | 1 | 52.39 | 1.08% | 243.08 | 243.08 | 112.72 | 0.000 |
| Square | 4 | 2674.39 | 55.00% | 2674.39 | 668.60 | 310.04 | 0.000 |
| A × A | 1 | 35.47 | 0.73% | 0.02 | 0.02 | 0.01 | 0.918 |
| B × B | 1 | 903.98 | 18.59% | 1016.97 | 1016.97 | 471.58 | 0.000 |
| C × C | 1 | 87.33 | 1.80% | 176.31 | 176.31 | 81.76 | 0.000 |
| D × D | 1 | 1647.60 | 33.89% | 1647.60 | 1647.60 | 764.02 | 0.000 |
| 2-Way Interaction | 6 | 1670.14 | 34.35% | 1670.14 | 278.36 | 129.08 | 0.000 |
| A × B | 1 | 55.50 | 1.14% | 55.50 | 55.50 | 25.74 | 0.000 |
| A × C | 1 | 430.15 | 8.85% | 430.15 | 430.15 | 199.47 | 0.000 |
| A × D | 1 | 366.72 | 7.54% | 366.72 | 366.72 | 170.05 | 0.000 |
| B × C | 1 | 2.40 | 0.05% | 2.40 | 2.40 | 1.11 | 0.307 |
| B × D | 1 | 162.56 | 3.34% | 162.56 | 162.56 | 75.38 | 0.000 |
| C × D | 1 | 652.80 | 13.43% | 652.80 | 652.80 | 302.71 | 0.000 |
| Error | 16 | 34.50 | 0.71% | 34.50 | 2.16 | | |
| Lack-of-Fit | 10 | 3.19 | 0.07% | 3.19 | 0.32 | 0.06 | 1.000 |
| Pure Error | 6 | 31.32 | 0.64% | 31.32 | 5.22 | | |
| Total | 30 | 4862.27 | 100.00% | | | | |
| **Model Summary** | | | | | | | |
| | S | R-sq | R-sq(adj) | PRESS | R-sq(pred) | | |
| | 1.46850 | 99.29% | 98.67% | 52.5538 | 98.92% | | |

In Figure 1b, a high F-value of 199.47 with a Prob > F of less than 0.00001 was found to be statistically significant for the interaction between (A) biomass solid loading (15% $w/v$) and (C) reaction time of pretreatment (30 min). Figure 1c shows an interaction plot between the factors (A) biomass solid loading (15% $w/v$) and (D) temperature (100 °C), which has a very high F-value of 170.05 with a Prob > F of less than 0.00001. That indicates a strong correlation between temperature and biomass under a highly acidic environment for the release of furfural in a medium with increased sugar release.

The response optimizer of the regression equation is statistically significant in minimizing the release of furfural in the process. Figure 1d displays an interaction plot between (B) H$_2$SO$_4$ concentration (12% $v/v$) and (C) reaction time of pretreatment (30 min). The plot shows a poor correlation with an F-value of 1.11 and a Prob > F of less than 0.307. This suggests that the interaction has no or negative impact on furfural release in rice straw hydrolysate, which is statistically insignificant.

Figure 1e depicts a response surface plot of the parameters (B) H$_2$SO$_4$ concentration (12% $v/v$) and (D) temperature (100 °C). The plot reveals a strong correlation with a high F-value of 75.38 and a Prob > F of less than 0.0001. This indicates that the parameters are statistically significant and contribute positively to minimizing furfural yield in pretreated

rice straw hydrolysate. Figure 1f illustrates an interaction plot between (C) reaction time of pretreatment (30 min) and (D) temperature (100 °C). The plot indicates that this interaction has the most profound impact on the release of furfural in pretreated rice straw hydrolysate in the pretreatment process. Furthermore, it has a very high F-value of 302.71 with a Prob > F of less than 0.0001. Therefore, this interaction has been identified as a key factor to be controlled to minimize furfural yield in the overall process.

The minimized release of furfural in the hydrolysate is highly influenced by a longer duration of treatment at high temperatures under an acidic environment. This process variable interaction was found to be the most statistically significant while maintaining the improved process efficiency of the conversion process of rice straw into precursor molecules as required for upfront into bioethanol fermentation steps. To validate the statistical results and the model equation, an analysis of variance (ANOVA) was performed and the findings are presented in Table 2. The ANOVA results demonstrate that the central composite design (CCD) model fits well and is highly significant, as indicated by the F value of 159.91 and a probability (Prob > F) of less than 0.0002. These statistical indicators reinforce the reliability and robustness of the model in predicting and optimizing the furfural release during the conversion process.

### 3.2. Statistically Optimized Model for Minimized Release of Hydroxymethyl Furfural (%) in Pretreated Rice Straw Hydrolysate

The regression equation for 5-hydroxymethylfurfural (HMF) (%) from the experimental trials is given below for the computation of HMF estimates.

$$\text{HMF (\%)} = -2.62 + 0.0285A + 0.1049B + 0.0684C + 0.0324\,D + 0.00106\,A \times A - 0.00199\,B \times B - 0.000927C \times C - 0.000027D \times D - 0.00495A \times B + 0.00336\,A \times C - 0.000953A \times D - 0.00028\,B \times C + 0.000257\,B \times D - 0.000488\,C \times D, \tag{2}$$

where Equation (2) predicts the percentage values of Hydroxymethyl furfural (%), in which A, B, C, and D are the coded values of biomass solid loading ($\%w/v$), $H_2SO_4$ ($\%v/v$), reaction time (min), and temperature (°C), respectively.

The response surface curve and interaction plots between different process variables for Hydroxymethyl furfural are shown in Figure 2a–f. The optimum level of pretreatment variables responsible for minimized Hydroxymethyl furfural (%) release in the rice straw hydrolysate was found to be in the range of, i.e., 0.03–2.32% Hydroxymethyl furfural (%)/100 g of solid pretreated biomass (Table 1).

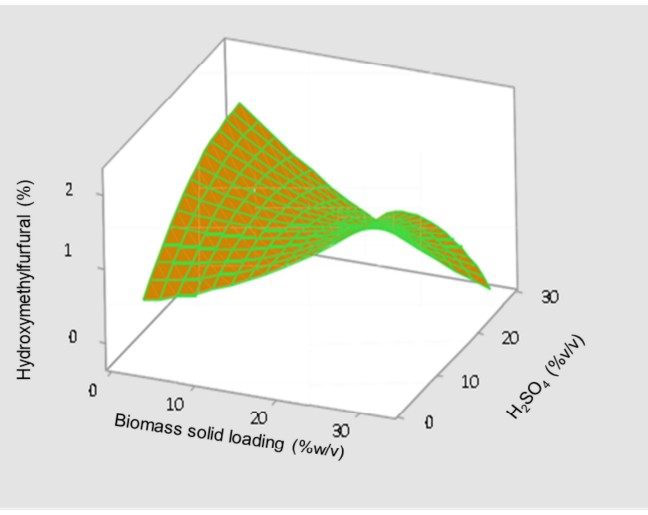

**a.** Surface plot of HMF (%) generation vs. (A) biomass solid loading %w/v vs. (B) $H_2SO_4$ (%v/v))

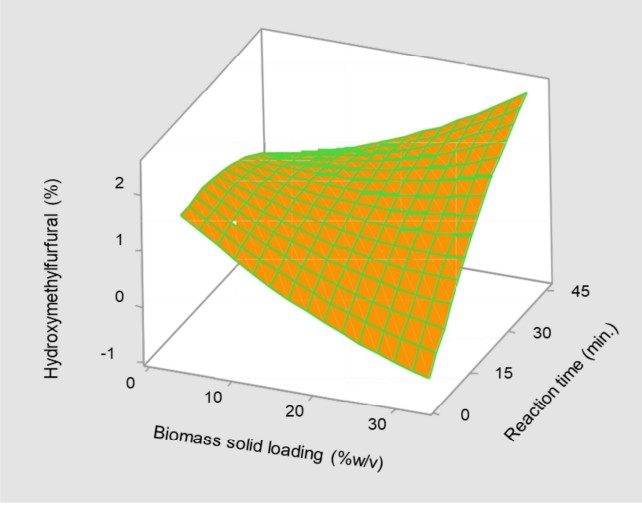

**b.** Surface plot of HMF (%) generation vs. (A) biomass solid loading (%w/v) vs (C) reaction time (min.)

**Figure 2.** *Cont.*

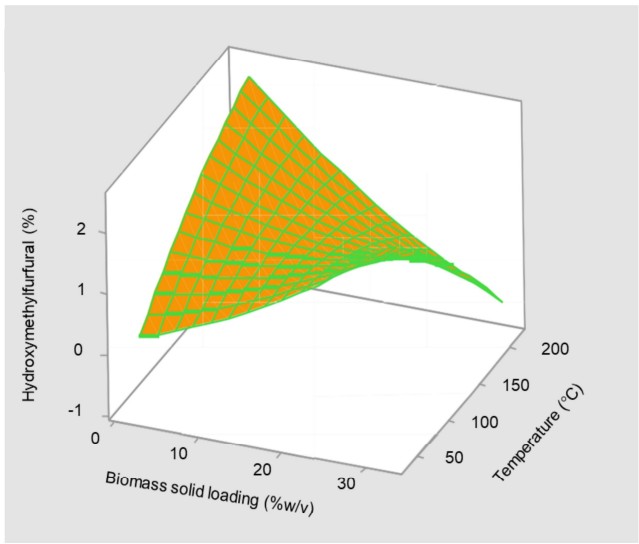

**c.** Surface plot of HMF (%) generation vs. (A) biomass solid loading (%w/v) vs. (D) temperature (℃)

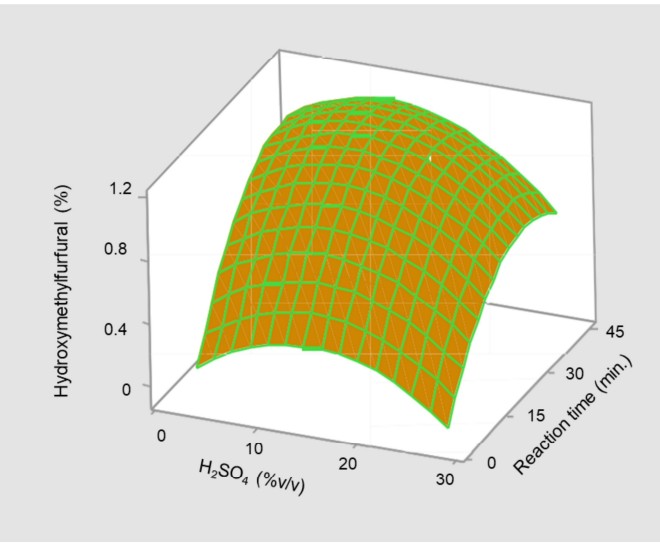

**d.** Surface plot of HMF (%) generation vs. (B) $H_2SO_4$ (%v/v), vs. (C) reaction time (min.)

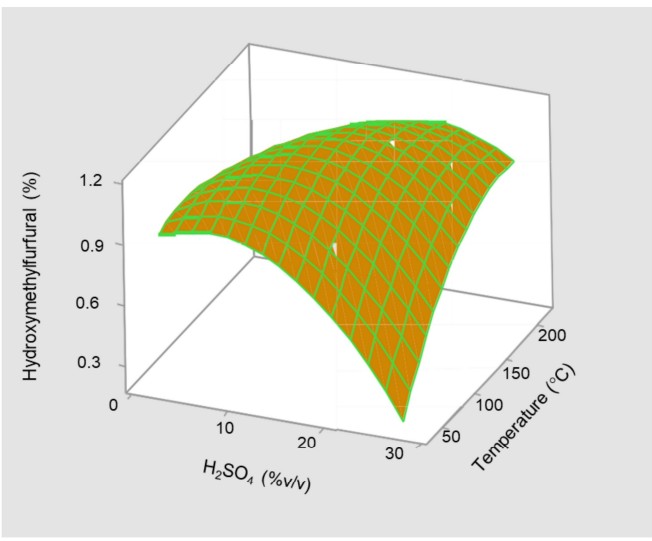

**e.** Surface plot of HMF (%) generation vs. (B) $H_2SO_4$ (%v/v) vs. (D) temperature (°C)

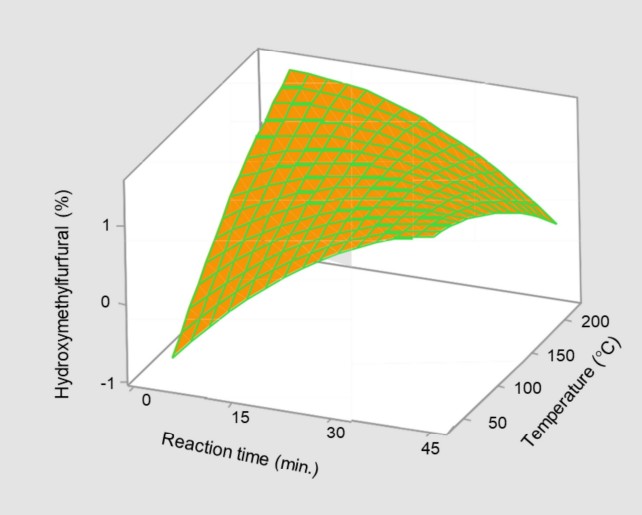

**f.** Surface plot of HMF (%) generation vs. (C) reaction time (min.) vs. (D) temperature (°C)

**Figure 2.** (**a–f**): The response surface curve and interaction plots between different process variables for Hydroxymethyl furfural generation.

The optimum values of each selected variable were found to be (A) biomass solid loading rate (15% *w/v*); (B) $H_2SO_4$ concentration (12% *v/v*); (C) reaction time of pretreatment is (30 min), and (D) temperature (100 °C). In a similar line of discussion, it may be observed from regression analysis as presented in Table 3 that the interaction between coded variable B vs. C in Figure 2d was found to have no or negative impact on the release of Hydroxymethyl furfural (%) in the pretreated rice straw hydrolysate with a very low F value of 0.05 with Prob < F of 0.833. All the interactions contributed generously to the release of Hydroxymethyl furfural in the hydrolysate and improved process efficiency.

**Table 3.** Response surface regression: HMF (%) vs. (A) biomass solid loading (%$w/v$), (B) $H_2SO_4$ (%$v/v$), (C) reaction time (min), (D) temp (°C).

| Source | DF | Seq SS | Contribution | Adj SS | Adj MS | F-Value | *p*-Value |
|---|---|---|---|---|---|---|---|
| Model | 14 | 7.5608 | 74.99% | 7.56077 | 0.54006 | 3.43 | 0.010 |
| Linear | 4 | 2.0651 | 20.48% | 0.20128 | 0.05032 | 0.32 | 0.861 |
| A | 1 | 1.3181 | 13.07% | 0.02624 | 0.02624 | 0.17 | 0.689 |
| B | 1 | 0.1126 | 1.12% | 0.00002 | 0.00002 | 0.00 | 0.991 |
| C | 1 | 0.6344 | 6.29% | 0.11612 | 0.11612 | 0.74 | 0.403 |
| D | 1 | 0.0000 | 0.00% | 0.02389 | 0.02389 | 0.15 | 0.702 |
| Square | 4 | 0.6827 | 6.77% | 0.68272 | 0.17068 | 1.08 | 0.398 |
| A × A | 1 | 0.1678 | 1.66% | 0.10312 | 0.10312 | 0.65 | 0.430 |
| B × B | 1 | 0.2072 | 2.06% | 0.23225 | 0.23225 | 1.47 | 0.242 |
| C × C | 1 | 0.2227 | 2.21% | 0.24820 | 0.24820 | 1.58 | 0.227 |
| D × D | 1 | 0.0850 | 0.84% | 0.08504 | 0.08504 | 0.54 | 0.473 |
| 2-Way Interaction | 6 | 4.8129 | 47.74% | 4.81291 | 0.80215 | 5.09 | 0.004 |
| A × B | 1 | 1.2390 | 12.29% | 1.23898 | 1.23898 | 7.86 | 0.013 |
| A × C | 1 | 1.0179 | 10.10% | 1.01789 | 1.01789 | 6.46 | 0.022 |
| A × D | 1 | 1.6561 | 16.43% | 1.65612 | 1.65612 | 10.51 | 0.005 |
| B × C | 1 | 0.0072 | 0.07% | 0.00724 | 0.00724 | 0.05 | 0.833 |
| B × D | 1 | 0.1202 | 1.19% | 0.12021 | 0.12021 | 0.76 | 0.395 |
| C × D | 1 | 0.7725 | 7.66% | 0.77247 | 0.77247 | 4.90 | 0.042 |
| Error | 16 | 2.5211 | 25.01% | 2.52112 | 0.15757 | | |
| Lack-of-Fit | 10 | 1.4732 | 14.61% | 1.47324 | 0.14732 | 0.84 | 0.614 |
| Pure Error | 6 | 1.0479 | 10.39% | 1.04788 | 0.17465 | | |
| Total | 30 | 10.0819 | 100.00% | 2.19% | | | |

| | | | Model Summary | | | | |
|---|---|---|---|---|---|---|---|
| | S | R-sq | R-sq(adj) | PRESS | R-sq(pred) | | |
| | 0.396951 | 74.99% | 53.11% | 9.86160 | | | |

### 3.3. Statistical Analysis of the Minimized Release of Total Phenolics (%) in Pretreated Rice Straw Hydrolysate

The regression equation for total phenolics (%) from the experimental trials is given below to compute total phenolics estimates.

$$\text{Total phenolics (\%)} = 1.160 - 0.0816\,A + 0.0111\,B - 0.0160\,C + 0.00902\,D + 0.000702\,A \times A - 0.002054\,B \times B - 0.000509\,C \times C - 0.000053\,D \times D - 0.000109\,A \times B + 0.000817\,A \times C + 0.000337\,A \times D + 0.002222\,B \times C - 0.000039\,B \times D + 0.000036\,C \times D, \tag{3}$$

where Equation (3) predicts the percentage values of total phenolics (%), in which A, B, C, and D are the coded values of biomass solid loading (%$w/v$), $H_2SO_4$ (%$v/v$), reaction time (min), and temperature (°C), respectively.

The response surface curve and interaction plots between process variables for total phenolics compounds release are shown in Figure 3a–f. The optimum level of pretreatment variables responsible for total phenolics compounds release in the pretreated rice straw hydrolysate was found to be in the range of, i.e., 0.478–1.647% total phenolics/100 g of solid pretreated biomass (Table 4). The optimal values for each selected variable were

determined to be: (A) biomass solid loading rate at 15% $w/v$, (B) $H_2SO_4$ concentration at 12% $v/v$, (C) reaction time of pretreatment at 30 min, and (D) temperature at 100 °C.

**Table 4.** Response surface regression: Total phenolics (%) versus (A) biomass solid loading (%$w/v$), (B) $H_2SO_4$ (%$v/v$), (C) reaction time (min), (D) temp (°C).

| Source | DF | Seq SS | Contribution | Adj SS | Adj MS | F-Value | *p*-Value |
|---|---|---|---|---|---|---|---|
| **Analysis of Variance** | | | | | | | |
| Model | 14 | 1.72940 | 72.76% | 1.72940 | 0.123528 | 3.05 | 0.018 |
| Linear | 4 | 0.31104 | 13.09% | 0.63483 | 0.158708 | 3.92 | 0.021 |
| A | 1 | 0.02419 | 1.02% | 0.00779 | 0.007787 | 0.19 | 0.667 |
| B | 1 | 0.03472 | 1.46% | 0.31259 | 0.312594 | 7.73 | 0.013 |
| C | 1 | 0.06202 | 2.61% | 0.01598 | 0.015984 | 0.40 | 0.539 |
| D | 1 | 0.19010 | 8.00% | 0.30289 | 0.302888 | 7.49 | 0.015 |
| Square | 4 | 0.69917 | 29.42% | 0.69917 | 0.174792 | 4.32 | 0.015 |
| A × A | 1 | 0.09360 | 3.94% | 0.04500 | 0.045001 | 1.11 | 0.307 |
| B × B | 1 | 0.21996 | 9.25% | 0.24796 | 0.247961 | 6.13 | 0.025 |
| C × C | 1 | 0.04729 | 1.99% | 0.07475 | 0.074752 | 1.85 | 0.193 |
| D × D | 1 | 0.33832 | 14.23% | 0.33832 | 0.338324 | 8.36 | 0.011 |
| 2-Way Interaction | 6 | 0.71919 | 30.26% | 0.71919 | 0.119865 | 2.96 | 0.039 |
| A × B | 1 | 0.00060 | 0.03% | 0.00060 | 0.000600 | 0.01 | 0.905 |
| A × C | 1 | 0.06003 | 2.53% | 0.06003 | 0.060025 | 1.48 | 0.241 |
| A × D | 1 | 0.20748 | 8.73% | 0.20748 | 0.207480 | 5.13 | 0.038 |
| B × C | 1 | 0.44422 | 18.69% | 0.44422 | 0.444222 | 10.98 | 0.004 |
| B × D | 1 | 0.00270 | 0.11% | 0.00270 | 0.002704 | 0.07 | 0.799 |
| C × D | 1 | 0.00416 | 0.18% | 0.00416 | 0.004160 | 0.10 | 0.753 |
| Error | 16 | 0.64739 | 27.24% | 0.64739 | 0.040462 | | |
| Lack-of-Fit | 10 | 0.18992 | 7.99% | 0.18992 | 0.018992 | 0.25 | 0.97 |
| Pure Error | 6 | 0.45747 | 19.25% | 0.45747 | 0.076245 | | |
| Total | 30 | 2.37679 | 100.00% | | | | |
| **Model Summary** | | | | | | | |
| | S | R-sq | R-sq(adj) | PRESS | R-sq(pred) | | |
| | 0.201151 | 72.76% | 48.93% | 1.68221 | 29.22% | | |

In Figure 3a, an interaction plot shows that the factors (A) biomass solid loading (15% $w/v$) and (B) $H_2SO_4$ concentration (12% $v/v$) had a statistically insignificant value of 0.01 with Prob < F of 0.905, indicating a negative interaction plot. In Figure 3b, a similar interaction plot shows that the factor (A) biomass solid loading (15% $w/v$) and (C) reaction time of pretreatment (30 min) had a statistically insignificant value of 1.48 with Prob < F of 0.241, indicating a negative interaction plot. Finally, in Figure 3c, the interaction plot between factors (A) biomass solid loading (15% $w/v$) and (D) temperature (100 °C) had a very low F value of 5.3 with Prob > F of less than 0.04, which is below the tested significance level of 0.05. Thus, the interaction in the plot was found to have the least or most minimal impact compared to other non-significant interactions.

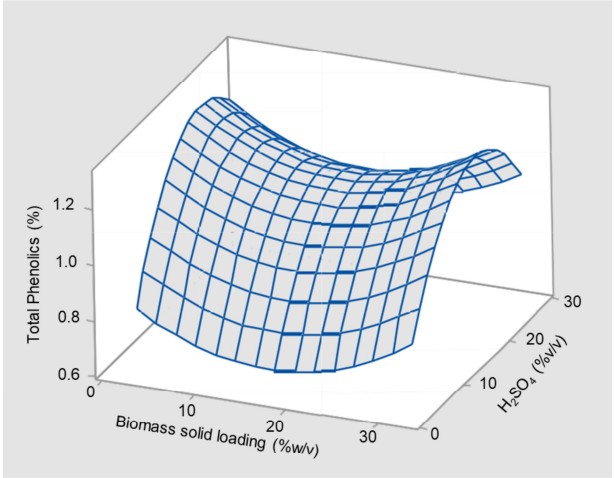

**a.** Surface plot of total phenolics generation (%) vs. (A) biomass solid loading (%w/v) vs. (B) $H_2SO_4$ (%v/v)

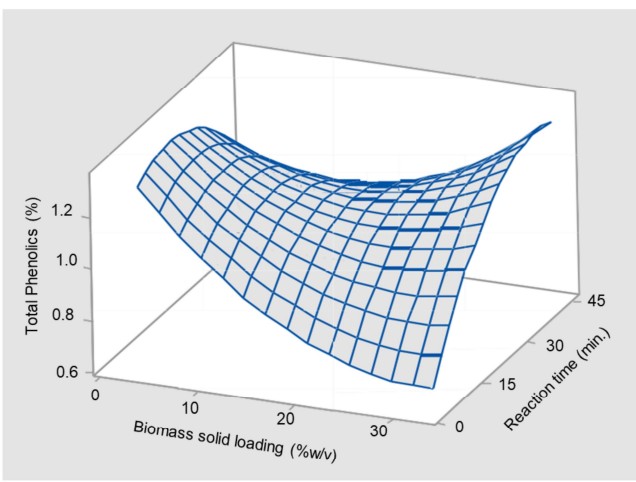

**b.** Surface plot of total phenolics generation (%) vs. (A) biomass solid loading (%w/v) vs (C) reaction time (min.)

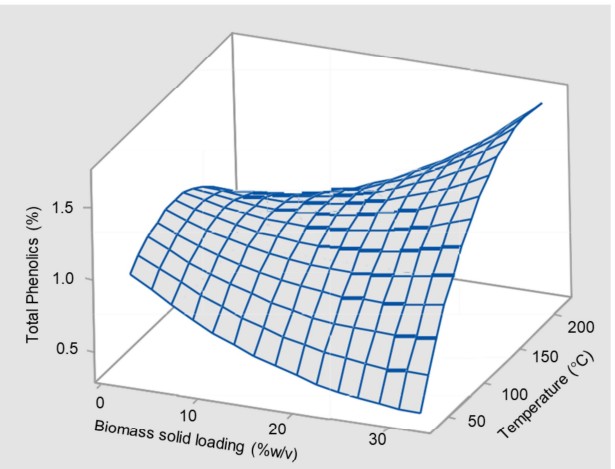

**c.** Surface plot of total phenolics generation (%) vs. (A) biomass solid loading (%w/v) vs. (D) temperature (°C)

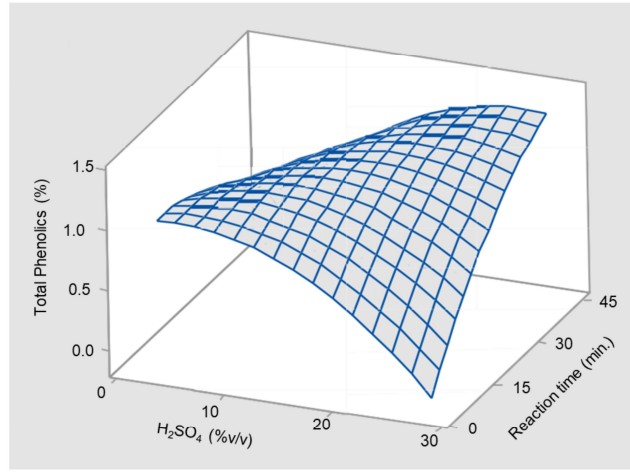

**d.** Surface plot of total phenolics generation (%) vs. (B) $H_2SO_4$ (%v/v), vs. (C) reaction time (min.)

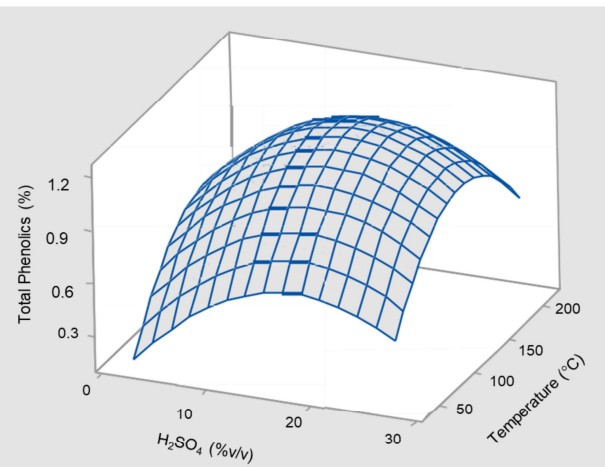

**e.** Surface plot of total phenolics generation (%) vs. (B) $H_2SO_4$ (%v/v) vs (D) temperature (°C)

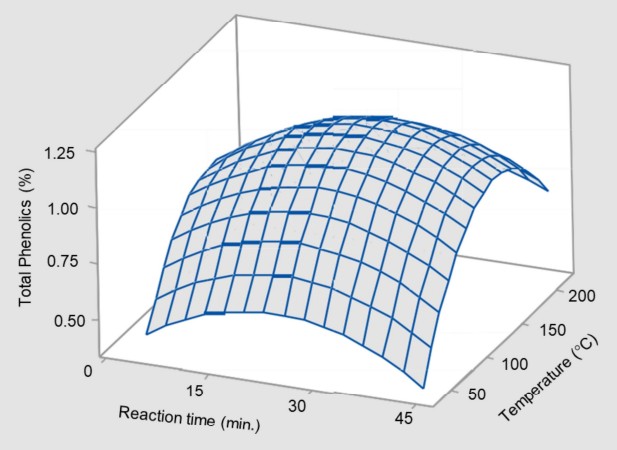

**f.** Surface plot of total phenolics generation (%) vs.(C) reaction time (min.) vs. (D) temperature (°C)

**Figure 3.** (**a–f**): The response surface curve and interaction plots between different process variables for total phenolics generation.

In Figure 3d–f, the interaction plots between (B) $H_2SO_4$ concentration (12% *v/v*) and (C) reaction time (30 min), (B) $H_2SO_4$ concentration (12% *v/v*) and (D) temperature (100 °C), and (C) reaction time (30 min) and (D) temperature (100 °C) were found to have

the least or no effect on total phenolics release in the hydrolysate. In Figure 3f, a surface plot is presented that shows the generation of total phenolics (%) vs. (C) reaction time (min.) vs. (D) temperature (°C). A negative impact on total phenolics generation in rice straw hydrolysate was observed at high biomass solid loading (15% $w/v$) and $H_2SO_4$ concentration (12% $v/v$). This result was validated by statistical Table 4, where a very low F value of 0.01 with Prob < F of 0.905 was computed, indicating a negative interaction plot. Validation of the statistical results and the model equation were analyzed using analysis of variance (ANOVA), presented in Table 3. The CCD model fits well, significantly showing the F value of 3.05 and Prob > F of less than 0.018.

The research findings were further analyzed using the normal probability plot of residuals to verify the assumption that the residuals are normally distributed. The study aimed to optimize the hydrolysis conditions of rice straw using the response surface methodology (RSM) to minimize the generation of inhibitors in the hydrolysate along with an improved process efficiency. The normal probability plot of the residuals was used to evaluate the goodness of fit of the RSM model by plotting the residuals against their expected values when the distribution is normal as shown in Figure 4a–c. The plot showed that the residuals approximately followed a straight line, indicating that the assumption of normal distribution was valid. This indicates that the RSM model used in the study was an appropriate tool for optimizing the hydrolysis conditions of rice straw.

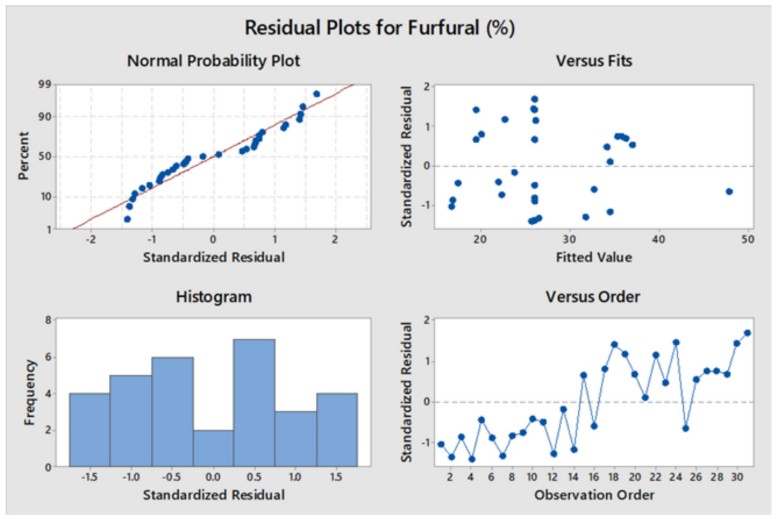

a. The normal probability plots of residuals for furfural

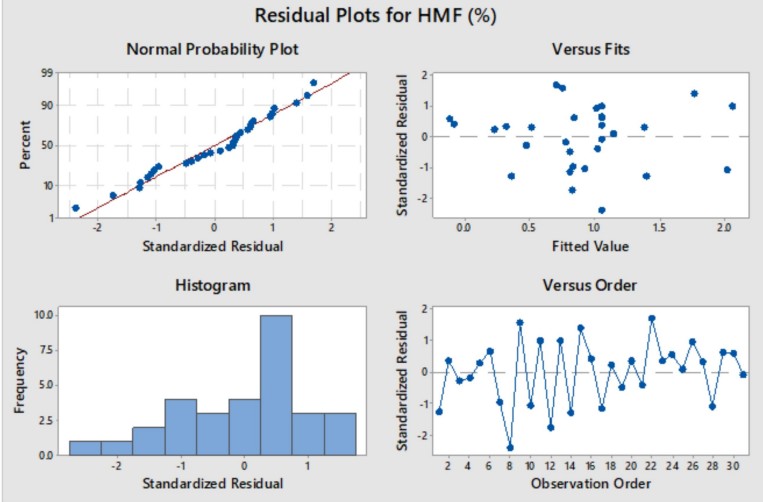

b. The normal probability plots of residuals for HMF

**Figure 4.** *Cont.*

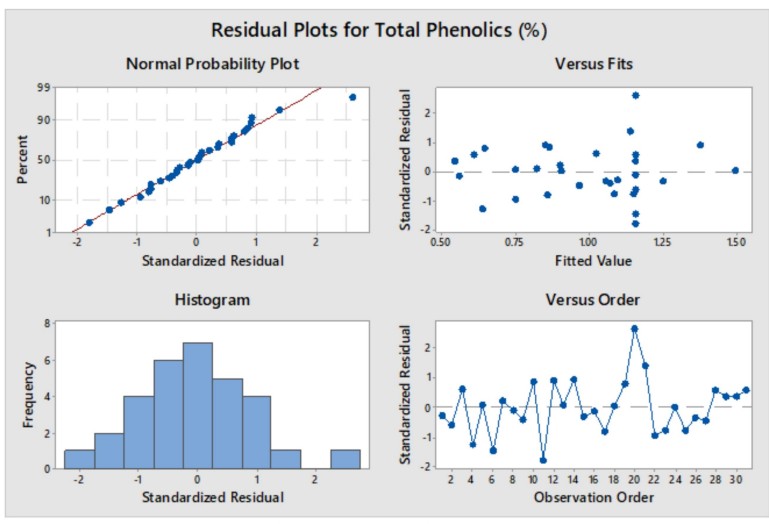

c. The normal probability plots of residuals for total phenolics

**Figure 4.** (**a**–**c**): The normal probability plot of residuals plots for (**a**) furfural, (**b**) HMF and, (**c**) total phenolics.

## 4. Discussion

This study conducted an analysis of the minimized release of furfural in the pretreated rice straw hydrolysate, which is a potential inhibitor molecule generated during the pretreatment of rice straw to obtain a higher yield of reducing sugar. The study found that the high temperature and acid concentration used in the pretreatment process resulted in the formation of furfural, 5-hydroxymethylfurfural, and total phenolics. The pretreatment condition that resulted in the reduced concentrations of these potential inhibitor molecules was found to be a biomass solid loading rate of 15% $w/v$, an $H_2SO_4$ concentration of 12% $v/v$, a reaction time of 30 min, and a temperature of 100 °C. We used a regression equation to compute the furfural estimates and validated the significance of the interaction between different process variables statistically. The response optimizer of the regression equation was found to be statistically significant in minimizing the release of furfural in the process. A strong correlation between temperature and biomass was found under a highly acidic environment for the release of furfural in a medium with increased sugar release. The interaction plot between (B) $H_2SO_4$ concentration (12% $v/v$) and (C) reaction time of pretreatment (30 min) was found to have no or a negative impact on furfural release in rice straw hydrolysate, which was statistically insignificant. On the other hand, the interaction between (C) reaction time of pretreatment (30 min) and (D) temperature (100 °C) had the most profound impact on the release of furfural in pretreated rice straw hydrolysate in the pretreatment process.

The study's findings provide valuable insights into the optimization of the rice straw pretreatment process to minimize the release of furfural in the hydrolysate. The reduced concentration of potential inhibitor molecules can improve the process efficiency of the rice straw chemical hydrolysis [31]. The research study conducted on rice straw hydrolysate also investigated the formation of other potential inhibitor molecules, namely 5-hydroxymethylfurfural (HMF) and total phenolics, during the pretreatment process. The study found that HMF was obtained in a concentration of 2.32%/100 g of solid biomass and total phenolics in a concentration of 1.65%/100 g of solid biomass, based on the treatment conditions presented in Table 1. HMF is a highly reactive molecule that is formed from the dehydration of hexoses during the pretreatment process. It has been reported to inhibit the growth of microorganisms, including those used in the production of biofuels and other bioproducts. In addition, HMF can react with other compounds present in the hydrolysate, leading to the formation of other toxic compounds. Therefore, minimizing the formation of HMF is critical for improving the process efficiency of rice straw hydrolysis [32]. Similar to furfural, the regression equation for HMF was also developed to predict the yield of HMF based on the process variables, i.e., biomass solid loading, $H_2SO_4$ concentration,

reaction time, and temperature. The results showed that the biomass solid loading, $H_2SO_4$ concentration, and temperature had a significant impact on the formation of HMF, while the reaction time had no significant effect on its formation.

The optimum operating conditions for minimizing the formation of HMF were found to be a biomass solid loading of 10% $w/v$, an $H_2SO_4$ concentration of 8% $v/v$, a reaction time of 30 min, and a temperature of 120 °C. Total phenolics are a group of compounds that are formed from the degradation of lignin and other plant cell wall components during the pretreatment process. These compounds have been reported to have antimicrobial activity and can also inhibit the fermentation process by inhibiting the growth of microorganisms. Therefore, minimizing the formation of total phenolics is important for improving the process efficiency of rice straw hydrolysis [33]. The regression equation for total phenolics was also developed to predict its formation based on the process variables. The results showed that the biomass solid loading, $H_2SO_4$ concentration, and temperature had a significant impact on the formation of total phenolics, while the reaction time had no significant effect. The optimum operating conditions for minimizing the formation of total phenolics were found to be a biomass solid loading of 15% $w/v$, an $H_2SO_4$ concentration of 10% $v/v$, a reaction time of 30 min, and a temperature of 100 °C. Overall, the study found that the parameters used to obtain a higher yield of reducing sugar in rice straw hydrolysate resulted in the formation of potential inhibitor molecules such as furfural, HMF, and total phenolics. The study demonstrated that optimizing the process variables could significantly reduce the formation of these potential inhibitor molecules, thereby improving the process efficiency of rice straw hydrolysis and increasing the yield of fermentable sugars for biofuel and other bioproduct production.

## 5. Conclusions

The primary concentration of potential inhibitors molecules obtained in pretreated rice straw hydrolysate was as follows: furfural (48.60%/100 g of solid biomass), 5-hydroxymethylfurfural (2.32%/100 g of solid biomass), and total phenolics (1.65%/100 g of solid biomass) with a pretreatment condition of (A) biomass solid loading rate (15% $w/v$), (B) $H_2SO_4$ concentration (12% $v/v$), (C) reaction time of pretreatment of 30 min, and (D) temperature found to be optimum at 100 °C. Optimizing these process variables reduced the inhibitor generation by up to one and a half times. The findings also suggest a need to remove generated degradation by-products in pretreated rice straw hydrolysate to further enhance the yield of total fermentable sugar in the rice straw hydrolysate and prevent the loss of fermentable sugar.

**Author Contributions:** Conceptualization, V.C.G., S.P. and M.S.; methodology, V.C.G., S.P. and M.S.; B.N.M.; investigation, V.C.G. and S.P.; writing—original draft preparation, V.C.G. and S.P.; writing—review and editing, V.C.G., S.P., M.S. and B.N.M.; visualization, V.C.G. and S.P.; supervision, M.S. and S.P.; project administration, S.P., M.S. and B.N.M. All authors have read and agreed to the published version of the manuscript.

**Funding:** This research was funded by the Dr. APJ Abdul Kalam Technical University (Lucknow) under collaborative research and innovation program (CRIP), grant number AKTU/Dean-PGSR/2019/CRIP/24.

**Institutional Review Board Statement:** Not applicable.

**Data Availability Statement:** Not applicable.

**Acknowledgments:** This research work is the part of Ph.D. Thesis of Vikas Chandra Gupta at IILM College of Engineering and Technology, Dr. A.P.J. Abdul Kalam Technical University (Lucknow). Vikas Chandra Gupta would like to thank to CST-UP students engineering research grant scheme and AKTU-CRIP faculty research grant scheme.

**Conflicts of Interest:** The authors declare no conflict of interest.

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
