# Peer review of "Minimization of Inhibitor Generation in Rice Straw Hydrolysate Using RSM Optimization Technique"

_agriculture, doi:10.3390/agriculture13071431_

Round 1
Reviewer 1 Report
The manuscript presents an interesting topic where Minimization of Inhibitor Generation in Rice Straw Hydrolysate Using RSM Optimization Technique. However, there are still some problems in this manuscript that need to be modified. Major revision is required before acceptance, as shown below.
1. There are many formatting errors in the manuscript, such as missing periods, improper use of parentheses, etc. Please check and revise it carefully, for example:
Page 1, line 12, the sentence “and HMF) are…”.
Page 1, line 25, the sentence “a temperature of 100οC…”.
Page 2, line 46, the sentence “inhibition problem [9] However, these methods…”.
Page 2, line 50, the sentence “acid [11], These inhibitors must…”.
Page 3, line 99, the first letter of this sentence “release [28]. the washed, dried and powdered rice straw…”should be capitalized.
Page 3, line 141-143, the format of these two sentences needs to be revised.
Please confirm and unify the format of “(%w/v)” and “(% v/v)” in the manuscript.
Page 4, “Temp (0c)” in the table heading.
Page 6, line 173, some punctuation in the legend needs to be removed or revised and the same check should be done for subsequent figures.
Page 8, line 218-219, the format of these two sentences needs to be revised.
Page 9, line 226, “reaction Time (Min), and Temp (0C), respectively.”.
Page 9, line 237, “and; (D) Temperature (100°C). in a similar line of discussion”.
Page 10, line 247-248, the format of these two sentences needs to be revised.
Page 11, line 267, “(B) H2SO4H2SO4 concentration (12% v/v) had a statistically”.
2. Page 1, line 24, please confirm and unify the format of “H2SO4” or “H2SO4” in the manuscript; “H2SO4concentration…” should also be revised.
3. Page 2, line 72-85, part of this paragraph is repeated, such as reference 24 and 27. It is suggested to simplify and merge and make a more concise summary.
4. Page 3, the references about the wavelength of substances such as furfural and HMF, and total phenolics compounds in the testing process need to be listed.
5. Page 6, line 176, please confirm and unify the format of “furfural” or “Furfural” in the manuscript.
6. Page 13, line 301, the Fig. 4 has multiple items that are unreadable and requires to be replotted.
7. Page 14, line 363, “5. Conclusions”, the number of the conclusion should be number “4”; line 369, the parentheses should be deleted in this sentence “treatment is (30 minutes) and (D) Temperature found to be optimum at (100°C).”
8. Which factor is most likely to reduce the production of inhibitors? Biomass solid loading, H2SO4 concentration, or others?
9. A suitable acid concentration combined with a long enough reaction time also seems to reduce the formation of inhibitors. Is it possible to reduce the cost of using high reaction temperature conditions instead?
10. Few references are outdated. If possible, and not essential to the argument and could be replaced with other.
11.According to Line 11 and Line 50, furfurals and HMF were inhibitors which were toxic to yeast cells, need to be controlled and reduced. But at Line 55, it is expressed that higher acid and temperature generated an improved chance of fermentable sugar yield. Are they conflicted?
12.-Page 2 Line 99 The first letter of the sentence was not capitalized.
13-The description of CCD (Line 102) and ANOVA (Line 214) was unclear in the section of Material and Methods.
14-What is the order of the experiments in Table 1? What is Std order?
15-How did the author obtain the conclusion of Page 4, Line 152-155 (Also in the conclusion section)? The conditions were not shown in Table 1.
16-Some Related works should be cited and may provide some insight for the author including:Co-combustion of Zn/Cd-hyperaccumulator and textile dyeing sludge: Heavy metal immobilizations, gas-to-ash behaviors, and their temperature and atmosphere dependencies. Chemical Engineering Journal 451 (2023) 138683; Fates of heavy metals, S, and P during co-combustion of textile dyeing sludge and cattle manure. Journal of Cleaner Production, 2023,383,135316;Converting and valorizing heavy metal-laden post-harvest hyperaccumulator (Pteris vittate L.) into biofuel via acid-pretreated pyrolysis and gasification. Chemical Engineering Journal,2023,467,143490.
Minor editing of English language required.
Author Response
Dear Reviewer, Thank you for your feedback. We appreciate your time and effort in reviewing our manuscript. We have carefully revised it to address following issues.

Reviewer 2 Report
The experimental article "Minimizing the formation of inhibitors in rice straw hydrolyzate using the RSM optimization method" is fully consistent with the topic of the publication "Agriculture". The relevance of the work is due to the use of a widespread raw material - rice straw and the search for solutions aimed at minimizing the formation of inhibitors in the rice straw hydrolyzate. It is known that in the process of chemical hydrolysis of rice salt, as well as other sources of plant origin, inhibitors are accumulated that adversely affect the growth and high level of yeast and the yield of the target product during the synthesis of bioethanol. This article expands the knowledge about the process of formation of inhibitors in terms of hydrolysis time and depending on the duration, temperature and sulfuric acid content. The article is well written and logically structured. Conclusions evaluate the objectives of the study.
I will give you 3 recommendations:
1) Bring the list of references in order with the requirements of the journal.
2) Table 1 seems very difficult. It is better to break it into several tables and organize the data.
3) Line 379: remove "Add" in "Funding".
Author Response

(The authors gave the same response as above.)

Round 2
Reviewer 1 Report
The question on point 16 was not fully answered and needs to be answered again. Check if this is the final modified version before proceeding. If so, I am quite disappointed.

Author Response
Response to Reviewer 1 Comments-Round-2
Reviewer’s comment: The question on point 16 was not fully answered and needs to be answered again. Check if this is the final modified version before proceeding. If so, I am quite disappointed.
“Point 16: Some Related works should be cited and may provide some insight for the author including:Co-combustion of Zn/Cd-hyperaccumulator and textile dyeing sludge: Heavy metal immobilizations, gas-to-ash behaviors, and their temperature and atmosphere dependencies. Chemical Engineering Journal 451 (2023) 138683ï¼› Fates of heavy metals, S, and P during co-combustion of textile dyeing sludge and cattle manure. Journal of Cleaner Production, 2023,383,135316ï¼›Converting and valorizing heavy metal-laden post-harvest hyperaccumulator (Pteris vittate L.) into biofuel via acid-pretreated pyrolysis and gasification. Chemical Engineering Journal,2023,467,143490. “
Response 16: By incorporating these pertinent citations, we aim to present a more robust and comprehensive analysis of our work, addressing the reviewer's comment effectively. We sincerely apologize for any disappointment caused and assure the reviewer that we have made significant improvements in this final modified version.
The following text added in the manuscript “The issue of utilizing potent lignocellulosic biomass, such as rice straw waste, through open-field burning, has emerged as a critical concern due to its negative impacts on greenhouse gas emissions and air pollution [1]. This situation poses significant threats to public health and the environment. However, we can mitigate the environmental consequences associated with the combustion of these agricultural biomass wastes by redirecting them towards green biorefinery. By implementing sustainable engineering concepts and cost-effective strategies, we can produce renewable energy, biofuels, organic chemicals, and building blocks for various polymer applications [2] This approach not only addresses the problem at hand but also promotes an environmentally friendly and efficient utilization of rice straw biomass, benefiting both the environment and society.
Furthermore, exploring innovative methods such as converting heavy metal-laden post-harvest hyperaccumulator (Pteris vittate L.) into biofuel via acid-pretreated pyrolysis and gasification can provide valuable insights for sustainable waste management and resource utilization [3]. By integrating these techniques, we can not only address the environmental impact of burning rice straw waste but also convert it into valuable resources, minimizing waste and promoting circular economy principles.” In the introduction part of the manuscript at page 1, line 30-45 and at page 2, line 46.
Also, we have updated all the references as per the reference style of journal in the references section of the manuscript.
